Screening frost-tolerant sunflower hybrids: integrating physiological traits and electrolyte leakage analysis

Kaya Mehmet Demir demirkaya76@hotmail.com 1
Kulan Engin Gökhan 1
Ergin Nurgül 2
1 Department of Field Crops/Faculty of Agriculture, Eskişehir Osmangazi University , Eskişehir , Türkiye
2 Department of Field Crops/Faculty of Agriculture and Natural Sciences, Bilecik Şeyh Edebali University , Bilecik , Türkiye
Zhang Qianwen
Electronic publication date: 2025 Nov 7
Publication date: 2025
Volume: 13
Electronic Location ID: e20282
Received 2024 Dec 24; Accepted 2025 Oct 1
Copyright: ©2025 Kaya et al.
Copyright year: 2025
Copyright holder: Kaya et al.
License: This is an open access article distributed under the terms of the Creative Commons Attribution License, which permits unrestricted use, distribution, reproduction and adaptation in any medium and for any purpose provided that it is properly attributed. For attribution, the original author(s), title, publication source (PeerJ) and either DOI or URL of the article must be cited.
License URL: https://creativecommons.org/licenses/by/4.0/

Keywords: Electrolyte leakage, Freezing damage, Growth stage, Helianthus annuus L., Selection

Funding: Scientific Research Projects Coordinatorship (BAP) of Eskişehir Osmangazi University No. 2020-23049 This work was supported by the Scientific Research Projects Coordinatorship (BAP) of Eskişehir Osmangazi University (Grant No. 2020-23049). The funders had no role in study design, data collection and analysis, decision to publish, or preparation of the manuscript.

==============================
Background

Frost is an important environmental stress factor that adversely affects plant growth and development and can even threaten plant survival.

Methods

This study aimed to identify frost-tolerant sunflower hybrids by analyzing the changes in physiological characteristics after exposure to frost at two early growth stages. Fourteen sunflower hybrids were exposed to frost stress at −4 °C for 4 hours at developmental stages of V2 and V4. Chlorophyll content (SPAD), relative leaf water content, leaf temperature, and electrolyte leakage were measured. Additionally, the percentage of damaged and dead plants following frost stress was recorded. Principal component analysis was performed for classification of sunflower hybrids.

Results

Significant differences were observed among the sunflower hybrids for all parameters investigated. All parameters were significantly affected by genotype and frost treatment. Increased electrolyte leakage and decreased relative water content were identified in frost- stressed plants. Electrolyte leakage was found to be significantly correlated with the percentage of damaged and dead plants at the V4 stage. Frost susceptibility increased with the advancing growth stage, with a higher percentage of damaged and dead plants at the four-leaf stage. The dead plant rate increased from 4.1% at the V2 stage to 19.4% at the V4 stage. SY Roseta, P63LE113, P64LE121, and Sanbro MR had the lowest percentage of damaged and dead plants at the two-leaf stage, while Sanbro MR, LG 50.585, and LG 59.580 were at the V4 stage.

Conclusions

The study highlighted the crucial influence of the growth stage on frost tolerance, with earlier stages showing greater resilience. Sunflower hybrids P63LE113, SY Gibraltar, and P63MM54 appeared more tolerant to frost damage. At the V4 stage, chlorophyll content and electrolyte leakage can be employed as potential selection criteria for frost-tolerant genetic resources and hybrids.

Introduction

Sunflower (Helianthus annuus L.) is the most important oilseed crop in Türkiye, accounting for 60% of the demand for high-quality vegetable oil (Kaya, Balalic & Milic, 2015). It is primarily cultivated in arid and semi-arid regions with a continental climate characterized by hot summers with low rainfall and cold, rainy winters under rainfed conditions (Kaya, Bayramin & Kaya, 2015). The adaptability and tolerance of sunflower to adverse soil and climatic conditions make it an important component of crop rotation. Because sunflower has been subjected to several environmental stressors, the supply of vegetable oil is closely related to the severity of abiotic stress factors such as drought and extreme temperatures encountered during the sunflower vegetation period (Gürbüz, Kaya & Demirtola, 2003).

Farmers prefer early sowing to mitigate the adverse effects of high temperatures and water scarcity during the vegetation periods of sunflower, particularly the flowering period. This can result in a reduction in seed yield and oil content. Although early sowing offers better water utilization in the soil and facilitates spring rains, it increases the risk of low temperatures during the initial growth stages (Gutierrez, Cantamutto & Poverene, 2016). In particular, temperatures below 0 °C (frost or freezing) interrupt plant growth, reduce productivity, and threaten the survival of sunflower plants by disrupting crucial metabolic processes (Hewezi et al., 2006; Hniličková et al., 2017). In Türkiye, frost damage occurs in agricultural areas almost every year. For example, spring crops such as maize, sunflower and sugar beet in the Marmara and Central Anatolia regions may be damaged by frost by late May between 2023 and 2025 (Turkish State Meterological Service (TSMS), 2025). This can lead to crop losses of up to 100%.

Frost injury occurs due to the crystallization of water within the cell (Beck et al., 2007). The production of ice crystals in the extracellular space enables water to leak from the protoplast into the extracellular space, leading to cell dehydration (Scott, 2000). Furthermore, rapid or repeated freeze–thaw cycles can cause the mechanical rupture of membranes, which severely impairs membrane integrity and ion homeostasis (Palta, Jensen & Li, 1982). Increased electrolyte leakage is a reliable indicator of frost-induced membrane damage and is frequently used to measure this phenomenon (Van Zuijlen et al., 2024). Frost impacts chlorophyll content (Gutierrez, Cantamutto & Poverene, 2016), electrolyte leakage (Tetreault et al., 2016), relative water content (Kulan & Kaya, 2024), protein synthesis (Duman & Wisniewski, 2014), osmotic potential (Hniličková et al., 2017), net carbon dioxide assimilation rate (Hejnák et al., 2014), antioxidant enzyme activities (Fabio, Tommasino & Grieu, 2022), photosynthesis (Liu, Qi & Li, 2012), and gene expression (Allinne et al., 2009). It is well-documented that genetic factors play a role in determining the frost tolerance levels of sunflower (Hniličková et al., 2017; Hejnák et al., 2014; Hernández et al., 2020); however, it is important to note that the tolerance of sunflower to frost stress may vary with the growth stage. Moreover, the duration of exposure and the temperature of frost may affect the tolerance levels of sunflower plants to frost stress. Therefore, this study was designed to identify the tolerance levels of sunflower hybrids according to growth stages, to determine the physiological characteristics of sunflower seedlings associated with frost tolerance at the V2 and V4 stages, and to clarify the genotypic variations among sunflower hybrids.

Materials & Methods

This study was conducted at the Seed Science and Technology Laboratory, Department of Field Crops, Faculty of Agriculture, Eskişehir Osmangazi University, Eskişehir, Türkiye, in 2021.

Plant material

Seeds of 14 sunflower hybrids (SY Gibraltar, SY Santos HO, SY Roseta, Sanbro MR, LG 5580, LG 50.585, LG 50.609SX, LG 59.580, P63LE113, P63LE119, P63MM54, P64LE121, Meteor CL and Duet CL) selected from different seed companies to create genetic variation and preferred extensively were used.

Plant growth conditions

After two days of pre-germination at 20 °C between moistened filter papers, the germinated seeds were transferred to plastic vials containing 30 cells (each measuring eight cm in diameter and 11 cm in height) for both the unstressed (control) and frost-stressed groups. A mixture of peat:perlite:vermiculite (3:1:1, v:v:v) was used as a plant growth medium. They were kept in the growth chamber for 2 days at 23/18 °C, 14/10 h night/day photoperiod with a relative humidity of 70% until the cotyledonary leaves emerged on the surface. On 30 April 2021, they were then transferred to their natural environment (open field conditions) under the protection of nets providing 50% shade, in order to obtain vigorous seedlings similar to those grown in field conditions. Irrigation was performed with tap water every other day according to the water requirements of the plants. Once the plants reached the desired growth stage, they were subjected to frost treatment.

Frost treatments

At the 2-leaf (V2) and 4-leaf (V4) growth stages, as defined by Schneiter & Miller (1981)), the plants were exposed to frost temperature at −4 °C for 4 h (Hniličková et al., 2017). The optimum combination of time and temperature for frost treatment was determined by pre-tests at −2, −4, −6, and −8 °C for 2, 4, 6, and 8 h. The differences among the sunflower hybrids with respect to damage and death rates were considered to determine the combination of time and temperature. The frost treatment applied to the plants in the growth chamber consisted of the following acclimatization conditions: 4 h at 10 °C, 4 h at 5 °C, 4 h at −4 °C, 3 h at 10 °C, and 3 h at 15 °C.

Measurements of physiological parameters

Leaf temperature and chlorophyll content

After taking out the plants from the incubator, they were left at room temperature for 4 h. The leaf temperature was then measured using an infrared thermometer (Trotec TP7 pyrometer), and the chlorophyll content was read using a Konica Minolta SPAD 502 chlorophyll meter as the SPAD value under laboratory conditions.

Relative water content

Fresh weights (FW) of the upper leaves of the plant were weighed and placed in distilled water for 24 h to determine turgor weights (TW). These leaves were dried in an oven at 80 °C for 24 h to determine dry weight (DW). The leaf relative water content (RWC) was calculated according to the formula described by Bijanzadeh, Barati & Egan. (2022).

RWC = [(FW–DW)/ (TW–DW)] × 100.

Electrolyte leakage

Electrolyte leakage was determined using six discs with a diameter of five mm that were excised from the true leaves. They were rinsed with distilled water and subsequently dried gently with paper towels. They were immersed in 20 mL of distilled water within 50 mL glass test tubes. The tubes were then incubated at 20 °C in darkness for 24 h, after which the electrolyte leakage was measured using an EC meter (WTW 3.15i) (EC1). Afterward, they were subjected to a 60-minute hot water treatment at 90 °C in a water bath to ensure complete tissue death, followed by a one-hour cooling period at room temperature prior to measuring ion leakage (EC2). The electrolyte leakage (EL) was calculated according to the methodology proposed by Lutts, Kinet & Bouharmont (1996).

EL (%) = (EC1/EC2) × 100.

Determination of damaged and dead plant rate

After the frost treatment, damaged plants were observed and counted. The plants were watered and allowed to grow in an incubator for another 3 days to determine the rate of dead plants.

Statistical analysis

This research was set up according to a two-factor factorial in a Completely Randomized Design and analyzed by computer using the MSTAT-C (Michigan State University, v2.10) program. Randomization was performed on sunflower hybrids. The significance level of the variations among sunflower hybrids was evaluated using Tukey’s test. The square root transformation was used to examine damaged and dead plant rates. Correlation analysis was performed to determine the relationship between damaged and dead plants and physiological characteristics. To rank sunflower hybrids for frost tolerance, principal component (PC) and cluster analysis were conducted using the JMP 13.0 program.

Results

Damaged and dead plants

There was a significant difference in the percentages of damaged and dead plants due to frost between sunflower hybrids at the V2 and V4 growth stages (Table 1). Data for the control treatment were not given in Table 1 because the control plants were not subjected to any frost treatment, resulting in no damage or death. A lower percentage of damaged and dead plants was observed at the V2 stage, and the results are presented in Table 1. As the growth stage progressed, the frost tolerance of the sunflower hybrids also decreased. The highest percentage of dead plants was recorded in Sy Santos HO, with 38.5% at V2, while no deaths were recorded at V4. Similarly, SY Gibraltar exhibited the highest percentage of dead plants (80.7%) at the V4 stage, but only 3.5% at V2.

Table 1 The percentage of damaged and dead plants of sunflower hybrids subjected to frost stress in the V2 and V4 growth stages.

Each value represents the mean of four replicates. Different letters within each column indicate statistically significant differences at the 5% level.

	V2 stage		V4 stage		
Hybrids	Damaged (%)	Dead (%)	Damaged (%)	Dead (%)	
SY Gibraltar	57.0ab	3.5bc	80.7a	73.3a†	
SY Roseta	3.5gh	0.0c	35.0bc	10.8de	
Sanbro MR	14.3efg	0.0c	4.3f	0.0e	
SY Santos HO	61.3a	38.5a	12.5ef	0.0e	
LG 50.585	46.5abc	0.0c	4.3f	0.0e	
LG 50.609SX	32.3b−e	7.0b	16.8de	12.5de	
LG 5580	42.8a−d	0.0c	30.0cd	30.0bc	
LG 59.580	27.3cde	0.0c	10.0ef	0.0e	
P63LE113	7.1h	0.0c	65.0ab	45.0ab	
P63MM54	21.8def	3.5bc	80.0a	45.0ab	
P64LE119	41.5a−d	0.0c	51.8abc	16.8cd	
P64LE121	7.2fgh	0.0c	32.5cd	10.0de	
Duet CL	40.5a−d	4.3bc	30.8cd	5.0de	
Meteor CL	24.3b−e	0.0c	66.8ab	23.3bc	
Mean	30.5	4.1	37.2	19.4	
Notes.

† Letters in each column indicate a significance level of 5%.

The present study showed that sunflower plants at the V2 stage tolerated frost stress better than plants at the V4 stage because a lower damaged and dead plant rate was obtained at the V2 stage. Younger sunflower plants were more tolerant to frost temperatures than older plants. The mean percentage of damaged plants increased from 30.4% at V2 to 37.7% at the V4 stage, leading to an increase in the dead plant rate.

Chlorophyll content and leaf temperature

Significant differences in chlorophyll content were observed among sunflower hybrids (Table 2). The minimum value (34.7 SPAD) was obtained from SY Santos HO, while the maximum (40.2 SPAD) was in LG 50.609SX. The chlorophyll content of sunflower hybrids responded differently to frost stress, increasing in six hybrids and decreasing in the rest. On the other hand, leaf temperature was reduced in plants affected by frost. The highest reduction was observed in SY Gibraltar, while the lowest was in Duet CL. In the present study, there were considerable changes between the sunflower hybrids for chlorophyll content at both V2 and V4 stages, but frost stress changed it considerably depending on the growth stage. The increase in chlorophyll content after frost was more pronounced at the V4 stage compared to the V2 stage. On the other hand, genotypic heterogeneity was very evident in chlorophyll content. For example, after frost treatment, SY Gibraltar increased chlorophyll content, but LG 50.609SX showed a significant drop. Furthermore, a positive and significant correlation was found between chlorophyll content and the percentage of damaged and dead plants at the V4 stage.

Table 2 Changes in chlorophyll content (SPAD) of sunflower hybrids subjected to frost stress at the V2 and V4 growth stages.

Each value represents the mean of four replicates. Differences indicate the change between frost-treated and control plants. Different letters within each growth stage indicate statistically significant differences at the 5% level.

Hybrid	V2 stage	Difference	V4 stage	Difference	
	Control	Frost		Control	Frost		
SY Gibraltar	38.0c−i	38.1c−h	+0.1	34.3l−o	40.6bc†	+6.3	
SY Roseta	37.2d−i	41.2ab	+4.0	35.2j−m	36.8g−j	+1.6	
Sanbro MR	35.6ijk	41.4a	+5.8	37.4e−i	34.2l−o	−3.2	
SY Santos HO	34.7jk	35.6ijk	+0.9	34.2m−o	34.3l−o	+0.1	
LG 50.585	38.7c−g	36.4g−j	−2.3	36.8g−j	35.8i−m	−1.0	
LG 50.609SX	40.2abc	37.5d−i	−2.7	45.5a	37.2f−i	−8.3	
LG 5580	36.8f−j	37.5d−i	+0.7	35.9i−l	39.0cde	+3.1	
LG 59.580	37.4d−i	37.1e−j	−0.3	36.3h−k	35.3j−m	−1.0	
P63LE113	37.8c−i	39.3a−e	+1.5	39.1cd	41.0b	+1.9	
P63MM54	38.9b−f	37.8c−i	−1.1	33.5no	41.2b	+6.9	
P64LE119	35.7h−k	30.2l	−5.5	33.4°	38.9def	+5.5	
P64LE121	39.6a−d	36.9e−j	−2.7	35.1k−n	37.9d−h	+2.8	
Duet CL	35.9h−k	40.0abc	+4.1	37.9d−h	36.6g−k	−1.3	
Meteor CL	36.8f−j	33.5k	−3.3	34.4l−o	38.1d−g	+3.7	
Mean	37.4	37.3		36.3b	37.6a		
Analysis of variance							
Hybrid (A)	**		**		
Stress (B)	ns		**		
AxB	**		**		
Notes.

ns, non-significant.

**, significant at 1%.

† Letters in each growth stage indicate a significance level of 5%.

At the V2 stage, leaf temperature decreased in frost-treated sunflower hybrids (Table 3). However, it increased in 7 of the sunflower hybrids and decreased in 5 hybrids. Duet CL and Meteor CL were the least affected hybrids by frost at the V2 stage. Leaf temperature was measured after frost stress, and lower values were recorded in the plants subjected to frost stress at the V2 stage. However, at the V4 stage, a significant difference was observed between sunflower hybrids, and increased leaf temperature was measured in SY Gibraltar, SY Roseta, and Sanbro MR after frost stress.

Table 3 Changes in leaf temperature (° C) of sunflower hybrids subjected to frost stress at the V2 and V4 growth stages.

Each value represents the mean of four replicates. The “Difference” columns indicate the change between frost-treated and control plants. Different letters within each growth stage indicate statistically significant differences at the 5% level.

Hybrid	V2 stage	Difference	V4 stage	Difference	
	Control	Frost		Control	Frost		
SY Gibraltar	24.1ab	18.9i	−5.2	19.2m	22.4fg†	+3.2	
SY Roseta	23.8a−d	19.3hi	−4.5	21.4hi	24.4c	+3.0	
Sanbro MR	23.5cd	19.2hi	−4.3	20.5jkl	23.8cd	+3.8	
SY Santos HO	23.3de	18.8i	−4.5	26.2b	26.2b	0.0	
LG 50.585	23.8a−d	19.4h	−4.4	20.8ijk	23.2e	+2.4	
LG 50.609SX	23.8abc	19.0hi	−4.8	26.9a	26.5ab	−0.3	
LG 5580	23.9abc	20.2g	−3.7	23.0ef	20.2kl	−2.8	
LG 59.580	23.7bcd	20.6g	−3.1	23.2e	20.6jkl	−2.6	
P63LE113	24.1ab	20.5g	−3.6	23.5de	20.8ijk	−2.7	
P63MM54	24.2a	21.8f	−2.4	24.4c	21.9gh	−2.5	
P64LE119	24.2a	21.6f	−2.6	21.0ij	20.9ij	−0.1	
P64LE121	24.1ab	21.9f	−2.2	20.6jkl	21.9gh	+1.3	
Duet CL	23.9abc	22.9e	−1.0	19.0m	20.1l	+0.2	
Meteor CL	23.5cd	21.9f	−1.6	20.5jkl	20.8jkl	+0.3	
Mean	23.8a	20.4b		22.1b	22.4a		
Analysis of variance							
Hybrid (A)	**		**		
Stress (B)	**		**		
AxB	**		**		
Notes.

** Significant at 5%.

† Letters in each growth stage indicate a significance level of 5%.

Relative water content and electrolyte leakage

The mean values of RWC showed that frost stress resulted in lower water content in sunflower hybrids (Table 4). The RWC in control and frost-stressed plants was significantly different. Differences in RWC between control and frost-stressed plants were negative and higher in SY Roseta, while P63LE113 produced higher RWC after frost stress. However, the cell membrane stability of sunflower hybrids increased in plants subjected to frost stress. The minimum difference was determined in LG.50.585, SY Roseta, and LG 59.580. There was great variation in the cell membrane stability of sunflower hybrids in control and frost-affected plants. At the V2 stage, it increased apparently in plants subjected to frost stress, but only one hybrid, Sanbro MR, showed a decrease at V4. Frost stress affected the relative water content of sunflower hybrids; however, significant variations were determined in unstressed plants. A significant change was observed in ten sunflower hybrids at both V2 and V4 stages.

Table 4 Changes in relative water content (%) of sunflower hybrids subjected to frost stress at the V2 and V4 growth stages.

Each value represents the mean of four replicates. The “Difference” columns indicate the change between frost-treated and control plants. Different letters within each growth stage indicate statistically significant differences at the 5% level. Asterisks indicate significance in the analysis of variance at the 5% (*) and 1% (**) levels.

Hybrids	V2 stage	Difference	V4 stage	Difference	
	Control	Frost					
SY Gibraltar	79.6d−g	79.8c−g	+0.2	87.8efg	88.8efg†	+1.0	
SY Roseta	86.2a	72.6lm	−13.6	96.2bcd	89.2efg	−7.0	
Sanbro MR	77.5e−j	79.4d−g	+1.9	97.0a	99.0bc	+2.0	
SY Santos HO	80.0c−g	73.0klm	−7.0	68.0m	75.0kl	+7.0	
LG 50.585	85.2ab	80.9cde	−4.3	75.6kl	95.8cd	+20.2	
LG 50.609SX	85.6ab	77.0f−j	−8.6	72.8lm	78.2jkl	+6.0	
LG 5580	79.9c−g	80.0c−g	+0.1	84.4ghi	84.4ghi	0.0	
LG 59.580	78.0e−i	78.3d−h	+0.3	89.1efg	84.0ghi	−5.1	
P63LE113	74.0j−m	83.6abc	+9.6	90.6def	91.5def	+0.9	
P63MM54	80.6c−f	74.3i−m	−6.3	85.8fgh	86.4fgh	+0.6	
P64LE119	82.0bcd	76.8f−k	−5.2	88.3efg	85.8fgh	−2.5	
P64LE121	76.7f−k	75.2h−l	−1.5	93.3cde	91.8def	−1.5	
Duet CL	77.2e−j	70.6m	−6.6	81.5hij	84.3ghi	+2.8	
Meteor CL	76.3g−l	79.7d−g	+3.4	78.8ijk	87.7efg	+8.9	
Ortalama	79.9a	77.2b		86.1b	88.0a		
Analysis of variance							
Hybrid (A)	**		**		
Stress (B)	**		*		
AxB	**		**		
Notes.

*, ** Significant at 5 and 1%, respectively

† Letters in each growth stage indicate a significance level of 5%.

The electrolyte leakage was found to be higher in sunflower plants at the V4 stage, and this was significantly correlated with the rate of damaged and dead plants (Table 5). Frost stress caused an increase in electrolyte leakage, but the differences between the control and frost-stressed plants were particularly evident at the V4 stage. The highest differences were observed in P64LE121, P63LE113, and SY Gibraltar at the V2 stage, while P63MM54, P63LE113, and Meteor CL exhibited the most notable variations at the V4 stage.

Table 5 Changes in electrolyte leakage (%) of sunflower hybrids subjected to frost stress at the V2 and V4 growth stages.

Each value represents the mean of four replicates. The “Difference” columns indicate the change between frost-treated and control plants. Different letters within each growth stage indicate statistically significant differences at the 5% level.

Hybrids	V2 stage	Difference	V4 stage	Difference	
	Control	Frost		Control	Frost		
SY Gibraltar	19.3fg	34.8ab	+15.5	49.7bc	49.9bc†	+0.2	
SY Roseta	33.1abc	34.5ab	+1.3	24.4j−m	37.1f	+12.7	
Sanbro MR	30.8c	36.4a	+5.6	24.6j−m	20.9m	−3.7	
SY Santos HO	20.9ef	32.0bc	+11.1	29.4hi	32.8gh	+3.4	
LG 50.585	29.9c	30.8c	+0.9	23.4klm	25.5i−l	+2.1	
LG 50.609SX	25.1d	34.5ab	+9.4	35.9fg	39.5ef	+3.6	
LG 5580	21.5ef	32.4bc	+10.9	27.3ijk	43.1e	+15.8	
LG 59.580	26.1d	30.6c	+4.5	29.5hi	32.1gh	+2.6	
P63LE113	15.0h	31.8bc	+16.8	22.1lm	54.9c	+32.8	
P63MM54	17.1gh	30.6c	+13.5	28.9hi	62.6b	+33.7	
P64LE119	21.6ef	30.2c	+8.6	40.6def	59.5a	+18.9	
P64LE121	15.1h	32.3bc	+17.2	27.7ij	47.6d	+19.9	
Duet CL	24.3de	32.4bc	+8.1	41.3de	58.9a	+17.6	
Meteor CL	25.2d	30.0c	+4.8	39.5efg	60.1a	+20.6	
Ortalama	23.7b	31.9a		31.7b	44.6a		
Analysis of variance							
Hybrid (A)	**		**		
Stress (B)	ns		**		
AxB	ns		**		
Notes.

ns, non-significant.

** Significant at 1%.

† Letters in each growth stage indicate a significance level of 5%.

Correlation and principal component analysis

Correlation coefficients and significance levels among the parameters investigated were given in Fig. 1. At the V2 stage, a significant and positive relationship was found between chlorophyll content and electrolyte leakage. No significant correlations were determined between the percentage of damaged and dead plants and physiological parameters. However, at the V4 stage, chlorophyll content showed significant correlations with the percentage of damaged plants (r = 0.891**) and dead plants (r = 0.881**). Furthermore, electrolyte leakage was significantly linked with the percentage of damaged plants (r = 0.829**).

Figure 1 Correlation coefficients between damage-dead percentages and the physiological traits at the V2 and V4 growth stages.

The correlation coefficients between frost-induced damage and dead percentages and selected physiological traits -leaf chlorophyll content (Chl), leaf temperature (LT), relative water content (RWC), and electrolyte leakage (EL)- at the V2 and V4 growth stages. Values indicate the strength and direction of correlations between traits. Note: *, **: significant at 5% and 1%, respectively; ns: non-significant.

Principal component and cluster analysis were performed on the mean value of each hybrid subjected to frost stress at the V2 and V4 stages. The results indicated that PC1 (34.8%) and PC2 (30.7%) accounted for a combined 65.5% of the data variability at V2, while PC1 (61.2%) and PC2 (19.4%) represented a total variation of 80.6% at V4 (Figs. 1 and 2). This suggests that the parameters investigated were more effective in explaining the frost tolerance at the V4 stage. Sy Santos was found to be a frost-tolerant sunflower hybrid at the V2 stage, with electrolyte leakage and chlorophyll content being the most determinative characteristics (Fig. 2). On the other hand, P63LE113, Sy Gibraltar, and P63MM54 showed better tolerance to frost stress compared to other sunflower hybrids (Fig. 3).

Figure 2 Classification of sunflower hybrids at the V2 stage for frost tolerance using principal component and hierarchical cluster analysis.

Different colors indicate cluster numbers (1: red, 2: green, 3: blue, and 4: purple) and PC indicates principal component.

Figure 3 Classification of sunflower hybrids at the V4 stage for frost tolerance using principal component and hierarchical cluster analysis.

Different colors indicate cluster numbers (1: red, 2: green, 3: blue, and 4: purple) and PC indicates principal component.

Discussion

Frost is an important environmental stress factor that can develop suddenly and poses vital risks for summer-season crops during their early growth stages. Its prevalence is steadily increasing as a consequence of global warming, which negatively impacts on the ability of plants to survive. Therefore, it is essential to prioritize breeding or selecting frost-tolerant cultivars to mitigate the adverse effects of frost stress. In this study, sunflower hybrids extensively cultivated in Türkiye were evaluated for their tolerance to frost stress at the V2 and V4 growth stages, when the plants are frequently exposed to frost. In contrast to previous studies, the plants were grown under natural conditions at the time of sowing, which was on April 16th, 2021, in order to simulate real climatic conditions and plant growth.

The findings of this study corroborate the existence of genetic variations among sunflower hybrids with regard to their tolerance to frost stress, although this is markedly influenced by the specific growth stage. Similar findings were reported by Hernández et al. (2020), who identified genetic differences in the survival of plants subjected to freezing stress at the four-leaf stage. The wild sunflower biotypes exhibited a higher survival rate than the cultivated ones. Additionally, Liu et al. (2011) reported high variability among cultivated maize genotypes, with a survival rate of 97%. In our study, sunflower hybrids appeared to be more sensitive to frost at the V4 stage, which is consistent with previous research.

Chlorophyll content of sunflower plants plays a key role in frost sensitivity, particularly in those at the V4 stage of development. Higher chlorophyll content may result in higher electrolyte leakage, damaged and dead plant rates. Similarly, most of the previous studies have indicated a reduction in chlorophyll content in the plants exposed to low temperatures (cold, chilling, and freezing) (Fabio, Tommasino & Grieu, 2022). The present study revealed that sunflower hybrids showed different responses to frost stress, with a marked variation in chlorophyll content. These differences were strongly correlated with the percentage of damaged and dead plants at the V4 stage (r = 0.891** and r = 0.881**, respectively), indicating that the level of frost tolerance in sunflower hybrids may be closely linked to their chlorophyll content.

Leaf temperature was a potential indicator for the plants subjected to various stress factors. It was lower in the sunflower plants subjected to frost temperature at the V2 stage. In a study conducted by Hirayama, Wada & Nemoto (2006), a significant positive relationship was identified between leaf temperature and rice yield. Furthermore, Perera, Cullen & Eckard (2019) found that this relationship was more reliable for the estimation of pasture production than air temperature. It is hypothesized that lower leaf temperature indicates that plants were not affected by frost treatment, as higher leaf temperature has been observed in plants subjected to stress factors. For example, Liu et al. (2011) demonstrated that higher leaf temperature was measured in maize plants subjected to water stress.

Frost stress changed the relative water content of sunflower hybrids, and the growth stage influenced it significantly. However, a stable increase or decrease in RWC was not observed due to frost stress. This result confirms the findings of Zareei et al. (2021), who determined a reduction in the RWC of strawberry seedlings exposed to decreasing freezing temperatures. Similarly, Samarina et al. (2020) found that the leaf RWC content of Camellia sinensis L. diminished by cold and frost temperatures, with a significant difference between the two cultivars. No significant changes in RWC were observed in cv. Gruzinskii7. In a recent study, Kulan & Kaya (2024) reported that the RWC content of sugar beet leaves exhibited variation according to the genotype and growth stage, and a notable increase was recorded at the V2.1 stage. In this study, a stable increase or decrease due to frost treatment was not observed among sunflower hybrids. This means that the RWC was not found to be a reliable parameter for the evaluation of sunflower genotypes under frost stress.

Electrolyte leakage was higher in plants subjected to frost stress at the V2 and V4 growth stages. Similarly, Hejnák et al. (2014) demonstrated that relative electrolyte leakage was higher in sunflower genotypes subjected to freezing stress. An increased level of electrolyte leakage is indicative of tissue damage caused by freezing. Fiebelkorn & Rahman (2016) determined that winter hardiness canola exhibited reduced water content and electrolyte leakage. A significant difference was identified between sunflower genotypes in response to cold stress, with electrolyte leakage being higher in cold-stressed plants. The sunflower hybrid Pampero exhibited greater electrolyte leakage than the Sierra hybrid (Fabio, Tommasino & Grieu, 2022). Our results also showed significant differences in electrolyte leakage between sunflower hybrids with varying degrees of frost tolerance. This result is corroborated by the findings of Hniličková et al. (2017), who identified genotypic differences in sunflower hybrids in response to freezing stress through electrolyte leakage assessment. Furthermore, significant correlations were observed between electrolyte leakage and the rate of damaged and dead plants at the V4 stage. However, at the V2 stage, increased electrolyte leakage was noted in P64LE121, P63LE113, and P63MM54, while P63MM54, P63LE113, and Meteor CL demonstrated a greater exclusion of electrolytes from leaves at the V4 stage.

PC and cluster analysis performed by mean values of the investigated parameters of sunflower plants at V2 and V4 stages subjected to frost stress such as damaged plant rate, dead plant rate, chlorophyll content, leaf temperature, relative water content, and electrolyte leakage revealed that four main groups were constituted. The sunflower hybrids responded differently to frost, and SY Santos was found to be more tolerant to frost stress at the V2 stage. However, the superiority of P63LE113, SY Gibraltar, and P63MM54 appeared at the V4 stage.

Conclusions

Frost often occurs in summer season plants, particularly those that are sown early, as this is required for high yields and quality under rainfed conditions. The impact of frost temperatures on plant growth and survival is mediated by disruptions of physiological and biochemical metabolic processes. However, the growth stage and genetic factors are of great importance in determining tolerance to frost stress. The present study demonstrated that sunflower plants at the V2 stage exhibited greater resilience to frost stress compared to those at the V4 stage, as evidenced by a lower percentage of damaged and dead plants. No significant relationship was identified between the parameters investigated and damaged or dead plants at V2. Conversely, at the V4 stage, rates of damaged and dead plants showed positive and significant correlations with chlorophyll content and electrolyte leakage. It can be argued that the V4 stage was suitable for frost-selection of sunflower genotypes. In conclusion, the selection of frost-tolerant genotypes in sunflower should be conducted at the V4 stage, with chlorophyll content and electrolyte leakage serving as valuable criteria for assessing frost stress. It is very important to select frost-tolerant sunflower hybrids for early sowing. This allows sunflower plants to grow taller and makes them more susceptible to frost. This result contributes to the sustainability of sunflower production in areas prone to early spring frosts, such as the Central and South Anatolia Regions of Türkiye.

Supplemental Information

Supplemental Information 1 Raw Data of V2-V4 periods

Each data point plotted indicates the average mean of 4 replicates of V2-V4 periods. All replicates are included.

Supplemental Information 2 Statistical analyses performed during V4 period

Supplemental Information 3 Statistical analyses performed during V2 period

Additional Information and Declarations

Competing Interests

Author Contributions

Data Availability

The authors declare there are no competing interests.

Mehmet Demir Kaya conceived and designed the experiments, performed the experiments, analyzed the data, authored or reviewed drafts of the article, and approved the final draft.

Engin Gökhan Kulan conceived and designed the experiments, performed the experiments, analyzed the data, prepared figures and/or tables, authored or reviewed drafts of the article, and approved the final draft.

Nurgül Ergin performed the experiments, prepared figures and/or tables, and approved the final draft.

The following information was supplied regarding data availability:

The raw measurements are available in the Supplementary File.

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
