# Peer review of "Screening frost-tolerant sunflower hybrids: integrating physiological traits and electrolyte leakage analysis"

_PeerJ, doi:10.7717/peerj.20282_

## Round 0.1 · original submission · Major Revisions

·

Basic reporting

The introduction is good but contains a large number of references. I suggest reducing it.

Experimental design

This is a factorial experiment with a Completely Randomized Design. The first factor is 14 sunflower hybrids, the second factor is temperature (-2, -4, -6, and -8 °C ), and the third factor is time (2, 4, 6, and 8 hours). Tables 2,3,4 and 5 show two factors. The first factor is hybrid (A), and the second factor is stress (B).

Validity of the findings

It has not been determined which sunflower hybrid was the best.

·

Basic reporting

The manuscript lacks basic structures of the research article, has poor organization, and has an insufficient background.

Experimental design

Comments
1. Line 1-2, Your title is too shallow and doesn’t represent the whole work. Also seems very poor. SO, I suggest you use: “Screening Frost-Tolerant Sunflower Hybrids: Integrating Physiochemical Traits and Electrolyte Leakage Analysis” or another better than this.

Abstract section,
2. Line 22, Is it necessary to put SPAD in the bracket?
3. Line 38, The subject stands for what?
4. Line 39, Please keep the chronological order of the keywords.

Introduction section
5. Your introduction is simple and lacks some information’s as global and national production and productivity of sunflower, sunflower breeding efforts in Turkey, frost damage loss, and frost stress coverage in Turkey.
6. Lines 50-54, it need citation/source.

Methodology section
7. You need to follow the standard research method for the research article. For example, you better have subheadings like description of planting area and material, then treatments/factors applied, data collected, data analysis, etc.
8. There should be full information about the experimental site/location, i.e., season, time, and other things.
9. Lines from 76-79, you need to give full information on the 14 sunflower hybrids used. Better to have a table to give full information about the materials.
10. The reason for choosing these 14 materials chosen?
11. Lines 79-90, it is the main part of the experiment (frost stress). So, you need to separate it from the plant material section.
12. Lines from 92-123 seem not structured. You need to give/add one main title, like data collected or traits recorded. Whatever the name you used, it should be scientific and clear.
13. Line 127, The type of design used in the experiment should not be in the statistical analysis part. It is not the appropriate place. Check it and rewrite it.

Results section
14. Please separate your results according to the data you recorded and try to address them accordingly. Otherwise, merged results could not be clear and scientific
15. Lines 133-137, it is not necessary to put these sentences. Better move it to the introduction or discussion section.
16. Lines 165-166, once again, I strongly suggest that you rewrite your results. Since you didn’t clearly write your result. Because the correlation result should be written separately. Additionally, indicate the correlation coefficient value. The tables are cited inconsistently, showing your carelessness in presenting your results as scientific results.

Discussion section
17. Overall, it is difficult to give further comments since the discussion doesn’t focus on the main results needing scientific elaboration supported by existing literature. Kindly focus on your main findings and insights. There are some points you need to see,
a. Which stage is more prone to frost?
b. Which traits are more influenced by frost at each stage?
c. What are your correlation results suggesting?
d. Based on the current findings and existing literature, what could be your insightful directions?
18. Lines 202-203, How and why can higher chlorophyll content result in higher EL, damaged, and dead rate? It is contradictory to the next sentence (Lines 208-210). Be careful with such speculations.
19. In the last paragraph of your discussion (Lines 244-248), It needs to be rewritten. What does the correlation result suggest?

Conclusions section
20. Lines 255-257 it is a bit confusing. You are saying that sunflower is more sensitive at the V4 stage than the V2 stage to frost stress based on your current result. This is hard to convince the science community since it is believed that the early stage is more sensitive to harsh environments and most other stresses.

Validity of the findings

Although the experiment/study was good, the authors were unable to show its novelty, and they poorly discussed and interpreted it.

---

## Round 0.2 · Major Revisions

·

Basic reporting

Comments on the title and Abstract section,
1. Lines 1-2, Your title is too shallow and doesn’t accurately represent the entire work. Also, it seems very poor. Therefore, I suggest using: “Screening Frost-Tolerant Sunflower Hybrids: Integrating Physiochemical Traits and Electrolyte Leakage Analysis” or another title more suitable.
2. Line 22, Is it necessary to put SPAD in the bracket?
3. Line 38, What does the subject stand for?
4. Line 39, Please keep the chronological order of the keywords.
Introduction section
5. Your introduction is simple and lacks some key information, such as global and national production and productivity of sunflowers, sunflower breeding efforts in Turkey, frost damage losses, and frost stress coverage in Turkey.
6. Lines 50-54 need citation/source.

Experimental design

Methodology section
7. You need to follow the standard research method for the research article. For example, you should have subheadings such as a description of the planting area and materials, treatments or factors applied, data collected, data analysis, etc.
8. There should be complete information about the experimental site/location, i.e. season, time, and other things.
9. Lines from 76-79, You need to give complete information on the 14 sunflower hybrids used. It's better to have a table to provide full information about the materials.
10.Why were these 14 materials chosen?
11. Lines from 79-90, it is the main part of the experiment (frost stress). So, you need to separate it from the plant material section.
12. Lines from 92-123, it seems not structured. You need to add a main title, such as 'Data Collected' or 'Traits Recorded'. Whatever the name you used, it should be scientific and clear.
13.Line 127, The type of design used in the experiment should not be in the statistical analysis part. It is not an appropriate place. Check it and rewrite it.

Validity of the findings

Result section
14. Please separate your results according to the data you recorded and try to address them accordingly. Otherwise, merged results could not be acceptable and scientific
15. Lines 133-137, It is not necessary to put these sentences. It would be better to move it to the introduction or discussion section.
16. Lines 165-166: I strongly suggest that you rewrite your results, as they were not clearly presented. Because the correlation result should be written separately. Additionally, indicate the correlation coefficient values. The tables are cited inconsistently, indicating a lack of care in presenting your results as scientific findings.
Discussion section
17. Overall, it isn't easy to provide further comments since the discussion doesn’t focus on the main results that require scientific elaboration supported by existing literature. Kindly focus on your main findings and insights. There are some points you need to see,
a. Which stage is more prone to frost?
b. Which traits are more influenced by frost at each stage?
c. What are your correlation results suggesting?
d. Based on the current findings and existing literature, what could be your insightful directions?
18.Lines 202-203, How and why can higher chlorophyll content result in higher EL, damaged and dead rates? This contradicts the next sentence (Lines 208-210). Be careful of such speculations.
19. In the last paragraph of your discussion (Lines 244-248), it needs to be rewritten. What does the correlation result suggest?
Conclusion section
20. Lines 255-257, It is a bit confusing. You are saying that sunflower is more sensitive at the V4 stage than the V2 stage to frost stress based on your current result. This is challenging to convince the science community, as it is widely believed that the early stage is more sensitive to harsh environments and other stresses.

Additional comments

Dear Although the title and its objectives are interesting, the way the authors present the results doesn't align with the scientific article writing and publication structures, as well as data presentation. Hence, it would be better if the authors can consider the critical comments provided and resubmit it.

Reviewer 3 ·

Basic reporting

1. The manuscript is written in generally clear English, but some typographic and formatting issues (e.g., symbols, spacing) require correction for professional presentation.
2. The abstract summarizes the findings but overgeneralizes the utility of chlorophyll content and electrolyte leakage without sufficient qualification.
3. The introduction is adequately structured but would benefit from a clearer statement of novelty and a well-defined research gap.
4. Figures and tables are relevant and mostly well labeled, though Figure 1 is oversimplified and does not effectively communicate the frost treatment protocol.
5. The term “physiochemical” is misused and should be corrected to “physiological and biochemical.”
6. More recent and relevant literature should be included to enhance the background and contextualize the findings within current research (e.g., post-2022 sources).

Experimental design

1. The use of 14 sunflower hybrids and two developmental stages is suitable for assessing genotypic variation, but replication across seasons or field environments is lacking.
2. The choice of -4 °C for 4 hours is justifiable but not thoroughly explained; data from preliminary trials should be presented or referenced.
3. While standard physiological traits (SPAD, RWC, EL, LT) are included, additional biochemical or molecular markers would enhance mechanistic insight.
4. Details on the electrolyte leakage assay lack standardization, particularly concerning tissue-to-solution ratios and incubation controls.
5. SPAD values are used without validation through extracted chlorophyll content, limiting the biochemical accuracy of the analysis.
6. Statistical analysis relies on standard ANOVA, but assumptions are not addressed, and the potential benefit of mixed-effects models is not explored.

Validity of the findings

1. The positive correlation between chlorophyll content and frost damage at the V4 stage contradicts common physiological expectations and should be interpreted more cautiously.
2. Relative water content produced inconsistent results and should not be promoted as a reliable frost-tolerance indicator without further evidence.
3. Significant trait-damage correlations at the V4 stage partially support the conclusions, but trait thresholds or predictive models are lacking.
4. The absence of cluster analysis or PCA limits the ability to identify hybrids with multi-trait tolerance profiles.
5. The study does not assess post-frost recovery or final plant performance, which is critical for evaluating agricultural relevance.
6. The reference to “natural growing conditions” is misleading, as the plants were grown in containers and subjected to controlled frost stress.

Additional comments

1. The study addresses a timely and agronomically relevant issue of frost stress during early growth stages in sunflower, and the inclusion of 14 hybrids provides valuable comparative data for breeding applications.
2. While the physiological traits examined are useful, integrating biochemical markers (e.g., proline, MDA, antioxidant enzyme activity) would enhance the mechanistic understanding of frost tolerance.
3. The interpretation of increased chlorophyll content correlating with higher damage at the V4 stage is unconventional and should be discussed more critically, possibly considering stress-induced pigment concentration effects or measurement artifacts.
4. Consider revising the conclusion to emphasize that chlorophyll content and electrolyte leakage were only significantly associated with damage at the V4 stage, not V2, and therefore cannot be generalized across all early stages.
5. The manuscript would benefit from improved figure quality (especially Figure 1) and table streamlining to facilitate better visual communication of results.
6. Discussing the potential application of the identified tolerant hybrids under projected climate scenarios with increased temperature variability would increase the study's broader significance.
7. Overall, this work lays a good foundation, but methodological expansion and refined analysis are needed to reach its full potential.

Reviewer 4 ·

Basic reporting

In the present study, the authors compared the frost tolerance of fourteen sunflower hybrids after exposure to -4°C for 4 hours at two growth stages. The results show the importance of the growth stage for frost tolerance and that electrolyte leakage and chlorophyll content can be used as selection criteria for frost tolerance. The study is interesting, but the results are not well described. The authors should analyze their results better and explain them clearly and in detail. The changes in the investigated parameter at the V2 and V4 stages should be described separately and then compared. In some sentences, it was not written at which stage the results were obtained. The results section should be rewritten.
The importance of investigating the effect of low temperatures on sunflower plants and the aim of the study are clearly described in the Introduction.
Abstract: Developmental stages should be described when first mentioned (line 21). Then abbreviations V2 and V4 can be used.
Materials and Methods: What is the light intensity during plant growth and exposure to -4°C? Where were the plants treated at low temperatures?
It was written that the frost treatments were at the 2-leaf and 4-leaf stages. Which leaves were used for measurements? It was written at line 109, “the upper leaves,” and at line 118, “true leaves”
Results
The results for damaged and dead plants should be better described. According to the results presented in Table 1, 7 sunflower hybrids were more damaged at stage V2, and the remaining 7 hybrids were more damaged at V4. However, the percentage of dead plants was higher at V4. Thus, the results on damaged and dead plants should be described separately. The sentence “A lower percentage of damaged and dead plants was observed at the V2 stage…” (line 146) is not entirely correct. It should be mentioned that for 3 hybrids (Sanbro MR, LG 50.585, LG 59.580) the percentage of dead plants was zero at both V2 and V4 stages.
Line 159 – The meaning of the sentence “The minimum value (34.7 SPAD) was obtained from SY Santos HO, while the maximum (40.2 SPAD) was in LG 50.609SX” is not clear. These values were obtained for the controls at the V2 stage.
Line 160 – “The chlorophyll content of sunflower hybrids responded differently to frost stress, increasing in six hybrids and decreasing in the rest” – At which stage? Which data were statistically compared? Did you compare statistically the data on the chlorophyll content of all control and frost-treated sunflower hybrids?
Line 162 – “The highest reduction was observed in SY Gibraltar, while the lowest was in Duet CL” – At which stage?
There is a problem in describing the results on chlorophyll content and leaf temperature. There are 3 sentences on chlorophyll content, then 2 sentences on leaf temperature, and again a description of changes in chlorophyll content.
Line 179 – “The RWC in control and frost-stressed plants was significantly different” – Too general; it is not true for all hybrids.
Line 180 – “Differences in RWC between control and frost-stressed plants were negative and higher in SY Roseta, while P63LE113 produced higher RWC after frost stress”- at which stage?
The changes in RWC were described only with 2 non-informative sentences. How could the significant enhancement of RWC after frost treatment of LG 50.585 at V4 be explained?
The word “ortalama” is left in Tables 4 and 5.
During the discussion, please check your results carefully. For example, it was written that “a stable increase or decrease in RWC was not observed due to frost stress”, which is not completely correct. As mentioned above in line 179, the opposite is written.
It was concluded that “the selection of frost-tolerant genotypes in sunflower should be conducted at the V4 stage, with chlorophyll content and electrolyte leakage serving as valuable criteria for assessing frost stress”. On this basis, which are the most frost-resistant sunflower hybrids among the 14 investigated?

Experimental design

no comment

Validity of the findings

no comment

Additional comments

I wrote all my comments and suggestions in the basic reporting.

---

## Round 0.3 · Minor Revisions

Hi Dear Authors, we're sorry for the long wait regarding your manuscript. We're still missing one reviewer's recommendation to accept it for publication. Please carefully respond to the comments from Reviewer 5, as this will help us reach a final decision. We hope to accept your manuscript for publication once you have fully addressed their feedback. Thanks.

Reviewer 3 ·

Basic reporting

-

Experimental design

-

Validity of the findings

-

·

Basic reporting

Abstract section
1. Lines 25–32: Consider including some representative data.
2. Line 38: What does “Subjects” refer to? It seems incomplete.

Introduction section
3. Line 43-74: Some specific data or references should be added to show how frost has affected sunflower yield or caused losses in Türkiye. The part about how frost affects plant physiology is a bit too brief; it would be better to include a short explanation of the mechanisms. Also, the last paragraph moves directly to the research aim. It would be better to outline the existing gaps or limitations in previous studies before presenting the aim of this research.

Experimental design

Methodology section
4. Line 78-79: You need to provide specific and important information about the planting location,
5. Line 81-84: Explain why these 14 sunflower hybrids were selected, and include details about these hybrids.

Validity of the findings

6. The Results section is disorganized. Please separate them.
7. Line 181: “Leaf temperature is associated with the yield of crops” should be moved to the Discussion section.

Discussion section
8. The logic isn’t clear, and it needs more support from existing literature. A lot of it just compares the results with other studies, but doesn’t explain why these things happened. It would be better to add more explanation and your insights.

References
9. Please carefully check all references; several references lack DOI information.

Additional comments

Dear Authors,
Your study objectives are interesting. However, the paper needs some improvements. In particular, the Results should be better organized, and the Discussion should provide more explanation for the findings rather than only comparing them with other studies. I kindly suggest that the authors address the comments and revise the manuscript for resubmission.

---

## Round 0.4 · Minor Revisions

Please address the comments from our section editor:

> There are several aspects of the methods that are unclear. First: 30 plants were transferred the control and stress environments. Do the authors mean 30 plants of each hybrid line? Please be explicit. If it is 30 plants of each hybrid line, do all 30 plants of a given line go into the same vial? Are there replicate vials for each line? I'm also confused by the dimensions of the vial...2 dimensions are given but these must be 3D. if they are round please indicate which of the 2 measurements is the diameter and which is the height.

> Second the methods state that the plants were moved from the chamber to their "natural environment" after 2 days, when the cotyledons had emerged. What is the "natural environment? Agirucltural field? If they were in there natural environment after day 2, how was the frost treatment applied in the natural environment?

> The optimum time and temperature for frost treatment was determined by pre-tests. What criteria was used to determine "optimum"? Maximum damage? Maximum variance between lines? Something else? +

> For leaf temperature the methods state that the plants were left at room temperature fro 4 h before measuring temperature. I assume the point of this test is to assay transpirational cooling? What were the light conditions?

> methods state completely randomized design. please explain what was randomized.

> Table 1: did any control plants show damage or death? If not, please state this in the results (no need to modify the table).

> abstract states
"Frost is the most important environmental stress factor ". By what critera? I suggest changing this to "Frost is an important..."

> Bibliography needs to be alphabetized by last name of first author.'

Thanks!

·

Basic reporting

No comment.

Experimental design

No comment.

Validity of the findings

No comment.

---

## Round 0.5 · accepted · Accept

Thanks for addressing all the comments made by reviewers and editors.